# Functionalised Anodised Aluminium Oxide as a Biocidal Agent

**DOI:** 10.3390/ijms23158327

**Published:** 2022-07-28

**Authors:** Mateusz Schabikowski, Magdalena Laskowska, Paweł Kowalczyk, Andrii Fedorchuk, Emma Szőri-Dorogházi, Zoltán Németh, Dominika Kuźma, Barbara Gawdzik, Aleksandra Wypych, Karol Kramkowski, Łukasz Laskowski

**Affiliations:** 1Institute of Nuclear Physics, Polish Academy of Sciences, PL-31342 Krakow, Poland; mateusz.schabikowski@ifj.edu.pl (M.S.); andrii.fedorchuk@ifj.edu.pl (A.F.); dominika.kuzma@ifj.edu.pl (D.K.); lukasz.laskowski@ifj.edu.pl (Ł.L.); 2Department of Animal Nutrition, The Kielanowski Institute of Animal Physiology and Nutrition, Polish Academy of Sciences, PL-05110 Jabłonna, Poland; 3Advanced Materials and Intelligent Technologies Higher Education and Industrial Cooperation Centre, University of Miskolc, H-3515 Miskolc, Hungary; emma.szdoroghazi@uni-miskolc.hu (E.S.-D.); nemeth.zoltan01@gmail.com (Z.N.); 4Institute of Chemistry, Jan Kochanowski University, PL-25406 Kielce, Poland; barbara.gawdzik@ujk.edu.pl; 5Centre for Modern Interdisciplinary Technologies, Nicolaus Copernicus University in Torun, PL-87100 Toruń, Poland; wypych@umk.pl; 6Department of Physical Chemistry, Medical University of Bialystok, PL-15089 Białystok, Poland; kkramk@wp.pl

**Keywords:** surface functionalization, Fpg glycosylase, anodic aluminium oxide, oxidative stress, bacterial *E. coli* strains, antibiotics

## Abstract

In this article, we describe the antimicrobial properties of a new composite based on anodic aluminium oxide (AAO) membranes containing propyl-copper-phosphonate units arranged at a predetermined density inside the AAO channels. The samples were prepared with four concentrations of copper ions and tested as antimicrobial drug on four different strains of *Escherichia coli* (K12, R2, R3 and R4). For comparison, the same strains were tested with three types of antibiotics using the minimal inhibitory concentration (MIC) and minimum bactericidal concentration (MBC) tests. Moreover, DNA was isolated from the analysed bacteria which was additionally digested with formamidopyrimidine-DNA glycosylase (Fpg) protein from the group of repair glycosases. These enzymes are markers of modified oxidised bases in nucleic acids produced during oxidative stress in cells. Preliminary cellular studies, MIC and MBC tests and digestion with Fpg protein after modification of bacterial DNA suggest that these compounds may have greater potential as antibacterial agents than antibiotics such as ciprofloxacin, bleomycin and cloxacillin. The described composites are highly specific for the analysed model *Escherichia coli* strains and may be used in the future as new substitutes for commonly used antibiotics in clinical and nosocomial infections in the progressing pandemic era. The results show much stronger antibacterial properties of the functionalised membranes on the action of bacterial membranes in comparison to the antibiotics in the Fpg digestion experiment. This is most likely due to the strong induction of oxidative stress in the cell through the breakdown of the analysed bacterial DNA. We have also observed that the intermolecular distances between the functional units play an important role for the antimicrobial properties of the used material. Hence, we utilised the idea of the 2D solvent to tailor them.

## 1. Introduction

During pandemic days, a substantial increase of the usage of antimicrobial nanomaterials is observed [1,2,3,4]. Many innovative materials are still at the designing stage [5]. For this reason, the usage of well-known materials, based on metals is back in the spotlight [6,7,8]. Metals, such as silver or copper, can be toxic to bacteria because of their antimicrobial activities even at exceptionally low concentrations. Moreover, unlike other antimicrobial agents, they are stable under conditions currently found in the industry allowing for their use as additives. Compounds based on those materials have been widely used as antimicrobial agents in agriculture, healthcare, general industry and many more [9,10,11,12,13]. Using metallic copper or silver in industrial applications involves several challenges associated with the nature of these materials. Silver is relatively expensive, while copper is susceptible to corrosion. Nevertheless, metal nanoparticles have become more and more popular. As it has been proven, such a form of silver or copper exhibits strong antimicrobial properties which increase with the decrease of their hydrodynamic size. Therefore, many research teams work on obtaining the smallest metal nanoparticles.

Most of the studies found in literature on the antimicrobial activities of metal nanostructures are restricted to strains of pathogens and rarely extend to non-pathogenic eco-friendly microorganisms [14]. Hence, the impact of environmental accumulation of nanoparticles, especially on microbiological communities residing in landfills or bodies of water, which are crucial to environmental recycling, are unknown. Therefore, it is a vital task to limit the migration of antimicrobial agents to the environment. Thus, instead of the aforementioned approach, we propose the development of functional antimicrobial units homogeneously distributed inside porous matrices dedicated to this specific use. This structure offers the highest efficiency of pathogens elimination (the smallest possible size of metal particle—an individual ion or individual molecules and different mechanisms of pathogens elimination), the largest active surface of the metal operating locally and with significantly limited depletion of an antimicrobial agent and strongly reduced possibility of metal migration outside the active area (active molecules chemically anchored to surface of the porous matrix).

The analysed K12 and R2–R4 strains of *Escherichia coli* are not only the dominant species of the human aerobic bacterial flora and a flora of various habitats in which people live, e.g., bathrooms, clinics and hospitals, but they can also temporarily colonise the oropharynx and skin of healthy people. However, apart from saprophytic strains (harmless to humans), there are also strains of *E. coli* which are pathogenic for humans and cause various forms of acute diarrhea. Infection usually occurs through contaminated food and water, as in the case with other bacterial diarrhea, and (less frequently) through indirect contact. In industrialised countries, pathogenic enteric strains of *Escherichia coli* are rarely a component of the intestinal flora of healthy people, therefore they are considered strictly pathogenic. When the appropriate amount of bacteria is ingested by a person susceptible to pathogenic infection, strains of *Escherichia coli* have the ability to cause inflammation of the small and or or large intestine. Adequate gastric acidity has a disinfecting effect and protects (to a certain extent) against infection, therefore people with low gastric acidity are particularly susceptible to infections with pathogenic *Escherichia coli* strains. The source of the infection is a sick person or a vector (except for STEC/EHEC strains where cattle is the source). Intestinal diseases caused by pathogenic strains of *E. coli* occur in the form of epidemics or, in rare cases, with an increase in the incidence in the summer months which is the rule for bacterial diarrhea.

Therefore, we study these strains with the use of the analysed amidoximes to investigate their etiology and the mechanism causing their resistance to many known and commonly used antibiotics. Despite numbers of literature reports on amidoxime’s pharmacological and biological properties, their antibacterial activities are still being rediscovered. Thus, there exist the need for additional research on their cytotoxic effect on selected hospital bacterial strains causing diseases associated with blood infections such as sepsis. The Cu-compounds still need to be further studied in vivo to better delineate the pharmacological potential of this class of substances.

Until now, such compounds have not been tested for biological activity against pathogenic *E. coli* strains, so there is a need to clarify their role [15,16,17,18,19,20,21,22,23,24,25].

AAO membranes were selected as the substrate for the functional groups. They are characterised by parallel cylindrical pores, which are ordered hexagonally, that are perpendicular to the surface. The diameters of pores that can be achieved range from 5 nm to 10 μm. The ease of control over the ordered array of defined pores is the main advantage of the membranes [26,27,28,29,30,31], especially in applications where ceramic material is beneficial. In such cases, AAO membranes are viable alternatives to silicon and polymer membranes. Moreover, they are relatively easy to manufacture, of low fabrication costs, biocompatible and with excellent thermal and chemical stability. AAO membranes can be produced by lithography [32], however, the most common method for AAO fabrication is the self-organised two-step anodisation of aluminium in acidic electrolytes [33,34,35]. This process forms a structure consisting of two layers: at the top, an aluminium oxide layer with hexagonally ordered pores and a continuous dielectric barrier layer at the bottom [36].

The non-toxic character of AAO matrices allows for their use in medical applications. They have been applied as an interface between bones and bone implants [37,38]. Similar attempts were undergone for dental implants [39]. The pores of AAO matrices were also used for storing drugs with controllable release [40,41,42] strongly depending on the depth of the pores and their diameter [43].

In this article, we propose a novel nanocomposite: the anodised aluminium oxide containing copper phosphonate functional units for biomedical application—the elimination of pathogenic organisms. The crucial factor, in this case, is the control of the distribution of functional units.

## 2. Materials and Methods

The nanomaterial presented here is based on the AAO matrices with the vertical pores of approximately 60 nm in diameter distributed regularly in hexagonal arrangement. The thickness of the matrices is around 125 μm. The material, including the interior of pores, is covered with functional units: propyl copper phosphonate distributed with the assumed density. The statistical distance between molecules is maintained by using spacers during grafting process [44,45]. The amount of the spacers (the spacer-functional group ratio) determine the distribution of the functional groups.

We tested materials with four densities of the functional groups:AAO Cu 4: anodised aluminium oxide with functional copper phosphonate units distributed with the highest possible density (no spacer units between functionalities),AAO Cu 3: anodised aluminium oxide with single spacer between functional copper phosphonate units (half the density of AAO Cu 4),AAO Cu 2: anodised aluminium oxide with six spacers between functional copper phosphonate units (almost seven times less copper than AAO Cu 4),AAO Cu 1: anodised aluminium oxide with twelve spacers between functional copper phosphonate units (almost thirteen times less copper than AAO Cu 4).

The schematic illustration of the nanocomposite can be seen in Figure 1.

The sample preparation consists of two steps: the fabrication of anodic aluminium oxide membranes and its precise functionalisation with molecules containing copper.

Since the procedure of surface functionalisation with assumed distribution is currently under patent procedure, this part of synthesis will hereby be omitted.

### 2.1. The Preparation of AAO Membranes

The porous aluminium oxide membranes were prepared by the anodisation [46] of aluminium foil (99%, SmartMembranes, Halle, Germany) with the use of a custom-built reactor. The foil was cut into squares (3 cm × 3 cm) and heated at 550 ${}^{\circ }$C for 12 h in nitrogen atmosphere in the Nabertherm LT5/13 oven (Germany). Next, they were cleaned sequentially with technical-grade acetone and ethanol for five minutes each in a ultrasonic cleaner. The aluminium plates were then washed with demineralized water and placed in the reactor made of polypropylene with a cathode in the form of a platinum mesh.

The bottom part of the reactor is made of brass through which, both, an electric contact with the sample and cooling are provided. All of the described procedures in the reactor were performed with cooling from the thermostat which was set to 4.5 ${}^{\circ }$C.

Prior to anodisation, the aluminium foil was electropolished to clean and smooth out its surface and to remove a residual Al${}_{2}$O${}_{3}$. The process was realised by applying 20 V for 15 min after 100 cm${}^{3}$ of an electropolishing mixture was added to the reactor. The mixture was prepared as follows: 25 cm${}^{3}$ of perchloric acid (HClO${}_{4}$, 60% pure p. a., Chempur, Piekary, Poland) was added to 75 cm${}^{3}$ of ethanol (C${}_{2}$H${}_{5}$OH, 96%, Chempur, Poland) and mixed for 10 min using a magnetic stirrer.

After electropolishing, the reactor was thoroughly washed and filled with 100 cm${}^{3}$ 0.3 M oxalic acid (C${}_{2}$H${}_{2}$O${}_{4}$× 2H${}_{2}$O, a standard solution, prepared from 99.5% pure reagent, Chempur, Poland). This preliminary anodisation was performed at 40 V for two hours.

Typically for ordered membranes made by anodisation, the first layer of the aluminium oxide was removed to obtain better arrangement of seeding spots on the surface. The removal was made as follows.

The plate was thoroughly rinsed with demineralised water and dried. The layer of Al${}_{2}$O${}_{3}$ was removed chemically in a solution prepared as follows: 12 cm${}^{3}$ of H${}_{3}$PO${}_{4}$ (85%, pure p. a., Chempur, Poland) was added to 88 cm${}^{3}$ of demineralised water. Then, 1.8 g of chromium (VI) oxide (pure p. a., Chempur, Poland) was added to the mixture which was then stirred for 10 min with a magnetic stirrer. The whole solution (approximately 100 cm${}^{3}$) was poured into the reactor which was placed on a hot plate set to 60 ${}^{\circ }$C. The chemical removal of Al${}_{2}$O${}_{3}$ was performed for two hours with occasional manual stirring by gently shaking the reactor.

The main anodisation was performed according to the same principle as the preliminary anodisation with the exception of its duration which set for 23 h 30 min.

To obtain a free-standing membrane, the Al substrates needs to be removed. In order to do this, the plate was removed from the reactor and the excess of aluminium was cut off. The plate was put into a crystalliser with approximately 200 cm${}^{3}$ of an etching solution. The solution was prepared by dissolving 60 g of CuCl${}_{2}$× 2H${}_{2}$O (99%, pure p. a., Chempur, Poland) in 200 cm${}^{3}$ of demineralised water. To dissolve the powder, the bottle was vigorously shaken and then the solution was stirred for 10 min with a magnetic stirrer.

The removal of aluminium was performed at room temperature for approximately 10 min. After the process, the plate was rinsed with water and placed into a crystalliser with approximately 50 cm${}^{3}$ of a 10% FeCl${}_{3}$× 6H${}_{2}$O solution (99%, pure p. a., Chempur, Poland) to remove the precipitated copper. This process was also performed at room temperature for 10 min. Finally, the last cleaning was performed in 19% HCl prepared by diluting 38% HCl (pure p. a., POCH, Gliwice, Poland) with demineralised water. A sample was immersed for one minute at room temperature.

The last step in the preparation of a free-standing AAO membrane with open pores is related to opening the pores from the substrate side. Approximately 150 cm${}^{3}$ of 5% H${}_{3}$PO${}_{4}$ (diluted from 85% H${}_{3}$PO${}_{4}$, pure p. a., Chempur, Poland) was poured into a polypropylene container. The membrane was gently placed on a sieve of the container with its top surface up so that only its bottom (substrate side) was in contact with the solution. When the pores are opened (which usually takes between 1–2 h and is indicated by droplets appearing on the surface of the sample), the sample was rinsed with demineralised water and dried at room temperature in vacuum.

Resulting AAO matrices have a form of disks with the diameter of 15 mm and 125 μm in thickness.

### 2.2. The Functionalisation

Due to the pending patent procedure, we cannot provide the functionalisation route in details.

The procedure was based on the grafting method with applied spacer units in order to keep the assumed statistical distances between functional molecules and thus, their final amount on the surface.

The spacer number is set by setting the molar proportions between precursors of functional units and spacer molecules. All syntheses were carried out in the protective atmosphere of tetrafluoromethane in order to avoid humidity. Before functionalisation the AAO matrices were dried under vacuum and stored in a container filled with a protective gas.

### 2.3. Characterisation Methods

The morphology of the membranes was analysed with the use of the scanning electron microscope (SEM) TESCAN VEGA3 (TESCAN, Brno, Czech Republic) and the transmission electron microscope (TEM) FEI Tecnai G2 20 X-TWIN equipped with LaB6 emission source and an FEI Eagle 2 K CCD camera (Thermo Fisher Scientific, Waltham, MA, USA). The TEM images were processed using the Gwyddion software [47]. The calculations of the pore size of AAO membranes were conducted in ImageJ 1.53q software [48].

In order to confirm the assumed number of functional units in the samples (thus, their relative distribution), we utilised differential pulse anodic stripping voltammetry (DPASV) [49,50]. The measurements were done in the three-electrode configuration at room temperature with the use of SP150 potentiostat/galvanostat (Biologic, Seyssinet-Pariset, France). We used a custom-built cell with a conductive fastening (gold) to immobilise the measured matrix at the bottom. The reference electrode was Ag/AgCl with saturated KCl as the electrolyte, while a counter electrode used was a platinum sheet. We applied following conditions of the experiment: pulse height of 2.5 mV, pulse width of 100 ms, step height of 5 mV, and step time of 500 ms.

The specific surface area was measured according to the Brunauer-Emmett-Teller (BET) method at 77 K based on nitrogen gas adsorption/desorption isotherms with the use of Quantachrome Autosorb iQ-MP-XR (Quantachrome Instruments/Anton Paar, Austria).

For the statistical analysis of the biological assays results we used ANOVA tests available under JMP software [51]. The error bars in the graphs represent standard deviation of the appropriate result.

## 3. Results and Discussion

### 3.1. Morphology

The fabricated membranes are of highly ordered pores within a frame of a single grain of aluminium (Figure 2). The channels of the pores are well-defined, parallel to each other and go through the whole length of the membrane. The average size of the inner diameter of pores, calculated from digitally measuring diameters in SEM pictures on a sample of approximately 900 measurements, is 60.7 ± 7.6 nm (Figure 3). The second anodisation of aluminium of 23 h 30 min resulted in membranes having approximately 125 μm in thickness. Thus, the average growth rate of the porous aluminium oxide for this specific procedure is approximately 5.32 μm/h or 88.3 nm/min (however, it must be taken into account the varying growth rate of the layer at different stages of anodisation).

Due to high porosity granting additional accessible surface, the measured specific surface area of the membranes is 30.8 m${}^{2}$/g. For a bulk material, this is a large value rivalling the specific surface area of agglomerated nanoparticles.

### 3.2. Differential Pulse Anodic Stripping Voltammetry

The differential pulse anodic stripping voltammetry can produce quasi-quantitative results since the surface area under the DPASV peak is proportional to the number of detected charge carriers. In this case, we are able to compare the relative number of copper ions in the samples. Assuming that each phosphoric acid unit would contain exactly one copper ion, the number of functional units would be equal to the number of detected copper cations. A similar procedure was presented in our earlier work [44].

In order to prove that the functional units are also located inside alumina pores, we measured a functionalised non-porous aluminium oxide (commercially available in the form of sheets from Merck, Germany). For this, we used the highest amount of functional groups with no spacers in order to obtain as many functional units as possible at the flat surface. The functionalisation procedure was the same as for the porous matrix, and the sample was named AO Cu 4. The DPASV spectra of the investigated samples are shown in Figure 4.

As a reference, we measured a pristine AAO matrix with no functional units. The material did not exhibit any peak (for this reason, we did not show it in Figure 4). Taking this into account, the obtained results confirm the presence of the increasing content of copper-containing groups in the matrices. There are significantly larger peaks for the porous samples than in the case of functionalised nonporous alumina layer (AO Cu 4). The stronger signal must derive from additional groups and since the conditions of the functionalisation were identical it means that the surface available in the pores was also functionalised. A closer look at the variation of peak areas versus the functionalisation level confirms the successful syntheses. The highest area under the DPASV can be observed for AAO Cu 4 according to our expectation. Such a sample posses the highest assumed number of phosphoric acid units. Almost two times lower peak area can be seen for AAO Cu 3. Indeed, such a matrix posses two times lower of the theoretical number of copper-containing groups. Roughly seven times lower peak is assigned for AAO Cu 2, again, according to synthesis assumption. A similar trend can be observed for the remaining sample. Still, lower concentrations of the copper-containing units (higher concentrations of spacer groups) can be connected with relatively high error. However, the trend seems to be evident. This indirectly proves that we can obtain assumed distributions of the functional units as a result of increasing intermolecular distances between copper-containing groups by increasing the number of spacers. This is an important conclusion regarding further analysis of antimicrobial properties of the material.

### 3.3. Antibacterial Properties

The obtained results depict that the all studied concentrations, AAO Cu 4-1, have an inhibitory effect on each studied bacterial model. Varied inhibitory activity was noted depending on the nature of the substituent in the aromatic ring of the tested compounds. The minimal inhibitory concentration values for each model *E. coli* R2–R4 and K12 strains were visible on all analysed microplates after the addition of the microbial growth index, resazurin [15,16,18,19,20,21,22,23,24,25,49,52].

On the first plate, where the K12 strain was analysed, the colour change was observed only at the dilution of 10${}^{-7}$, and the MIC values of the compounds shown in (Figure 5) were calculated to be the concentration of 0.028 μM. On the second plate, on which strain R2 was present, the colour change was observed already at the dilution of 10${}^{-3}$, and the resulting MIC values were calculated at the concentration of 0.015 μMs. For the R3 and R4 strains, the colour change of the analysed compounds was visible already at the dilution of 10${}^{-2}$ and the corresponding MIC values were calculated at the concentration of 0.0025 and 0.02 μM for R3 and R4, respectively.

Similar values were observed for the minimum bactericidal concentration test (Figure 6a). Increasing MBC values were observed for all four analysed concentrations but at different levels. Bacterial strains R3 and R4 were more sensitive compared to K12 and R2 to the analysed compounds in both types of MIC and MBC assays. Strain R4 was the most sensitive of all strains, possibly due to the longer length of the lipopolysaccharide (LPS) chain. In all studied cases, the MBC values were approximately 35 times larger than the MIC values (Figure 6b). The modification of functional groups in the analysed Cu-compounds significantly changes the MBC/MIC ratio (Figure 6c) and strongly depends on specific concentrations which is clearly visible on the basis of the obtained data (Figure 5 and Figure 6 and Table 1).

The obtained MIC values indicate that the toxicity of the tested compounds against the analysed model bacterial strains K12 and R2–R4 should arise with the increase of the number of aromatic rings in peptidomimetics and with the appropriate length of the alkyl chain along with the specific type of substituent. Among all of the analysed compounds, the highest toxicity was observed for compounds marked as AAO Cu 2. Therefore, the membranes with this specific concentration of copper were selected for further studies on the basis of MIC and MBC values. They were used to modify model *E. coli* strains and additionally were digested with Fpg protein from the group of repair glycosylases which is a marker of oxidative stress.

As in previous studies on various types of compounds, we also wanted to observe the effect of modification on the magnitude of oxidative damage, which should be seen as extending strands compared to unmodified forms having the three forms ccc, oc and linear. The results of bacterial DNA modified with Cu-compounds (Figure 7 with the action of Fpg) showed that all analysed compounds can strongly change the topological forms of plasmids, even after digestion with the Fpg protein and are new substrates for them.

The changes in the main topological forms of the plasmid: ccc, oc and linear were observed in DNA isolated from model strains and digested with Fpg protein. About 4% of oxidative damage was identified after digestion with the Fpg protein. This indicates that the analysed compounds damage bacterial DNA very strongly due to the oxidative stress induced by them in the cell (Figure 7).

Our observations indicate that the Cu-compounds may determine the toxicity to some *E. coli* R4 strains, as evidenced by the MIC and MBC values. The obtained results were also statistically significant at the level of *p* < 0.05. (Table 1). Based on that, we conclude that the studied compounds can potentially be used as substitutes for commonly applied antibiotics (Figure 6a,b, Figure 8 and Figure 9).

We have not observed significant changes in topological forms in the analysed bacterial strains after DNA isolation from them, after the modification with antibiotics and digestion with Fpg protein and MIC analysis (Figure 7). This suggests that modifications with antibiotics are less recognisable by the Fpg protein than the modifications introduced into the bacterial DNA by the analysed composites. Most likely, the modifications of the antibiotics used in the bacterial DNA do not cause significant changes in the topological principles caused by the specific bacterial glycosylase to which the Fpg protein belongs. Large standard deviations in the K12 *E. coli* strain, which is a natural component of the human intestinal flora, result from differences in 13 specific places in the DNA base sequence where there are specific insertions or deletions, and inversions between the rrnD and rrnE ribosomal RNA genes [17].

On the other hand, significant modifications of plasmid DNA were observed for all of the four types of composites. Modifications with antibiotics were smaller and not as clear as in the case of the analysed Cu-compounds. The sensitivity of *E. coli* strains to the cytotoxic effect of the used compounds and after Fpg protein digestion were as follows: R4 > R2 > R3 > K12. This effect was similar to our previous studies [15,16,18,19,20,21,22,23,24,25,49,52].

It indicates very high cytotoxicity of the analysed amidoxime derivatives towards bacterial DNA. Most likely, it is the result of the modification of the components of the bacterial membrane and the LPS contained in it which may induce specific enzymes from the group of topoisomerases and helicases destabilising the structure of the exposed DNA bases. A stabilisation of the complex that regulates these enzymes is perhaps necessary for cell survival. Blocking these enzymes inhibits DNA replication and rewriting which can affect its total amount.

## 4. Conclusions

The obtained results revealed that carefully designed amidoxime derivatives may constitute a new potential source of innovative cheap substitutes for antibiotics against various types of bacterial microorganisms. We focused on the structure-activity relationship of the four compounds with a Cu scaffold. The obtained results showed a strong influence of the activity of all analysed concentrations on the values of MIC and MBC as well as MBC/MIC for various strains of *E. coli*: R2–R4 and K12. The above results are very important for research on the mechanism of cytotoxic action of new compounds as innovative and safe drugs based on amidoxime derivatives which may lead to the destruction of the bacterial cell membrane by changing its surface charge and may play a significant role in changing its electrokinetic potential expressing with the reversal of loads. The reported compounds may be highly specific for pathogenic *E. coli* strains on the basis of the used model strains.

In the future, cytotoxicity studies will also be conducted using various cell lines and cultures to assess the biocompatibility of test compounds for active peptidomimetics.

## Figures and Tables

**Figure 1 ijms-23-08327-f001:**
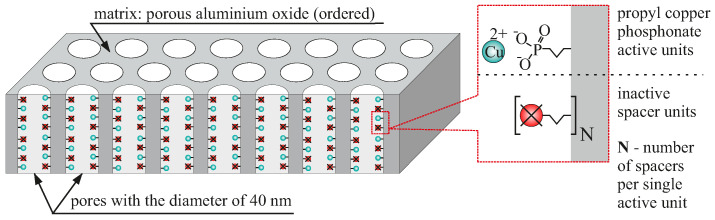
The schematic representation of the investigated material. The density of functional units is determined by the statistical number of spacer units (N) between functional molecules.

**Figure 2 ijms-23-08327-f002:**
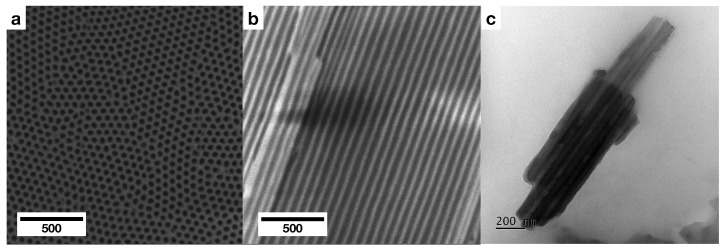
The SEM and TEM micrographs of the anodic aluminium oxide membranes: (**a**) a planar view, (**b**) a cross-section, (**c**) a transmission micrograph of a fragment of a cross-section of an AAO membrane.

**Figure 3 ijms-23-08327-f003:**
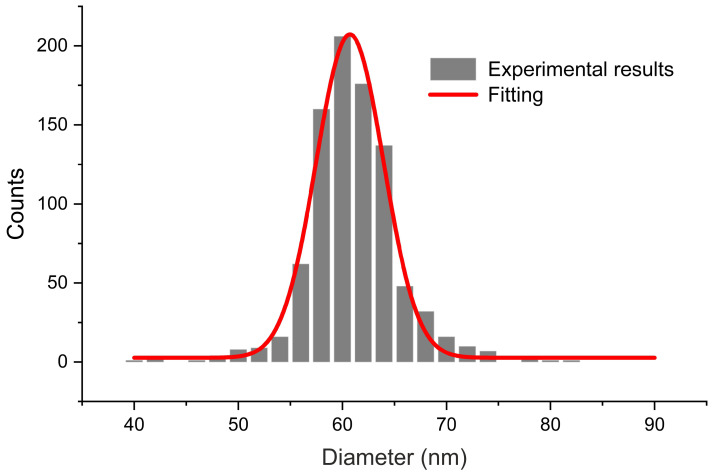
The pore size distribution of the AAO pores.

**Figure 4 ijms-23-08327-f004:**
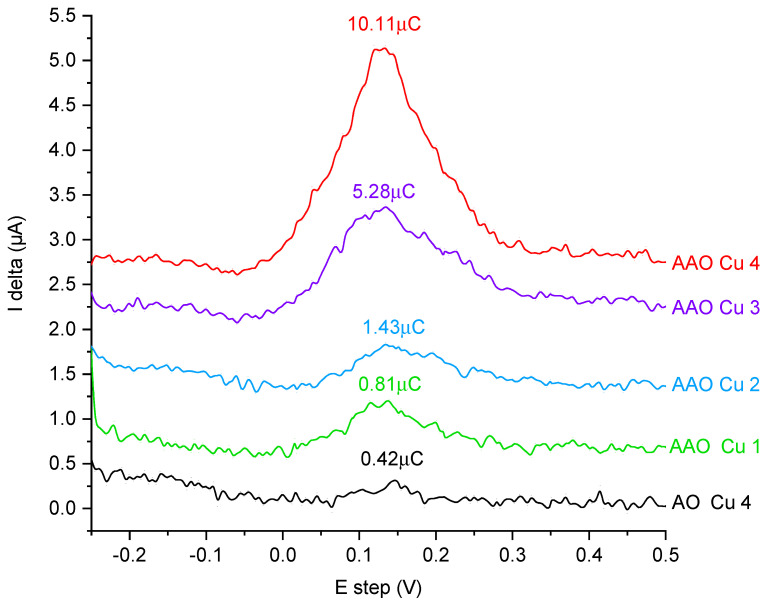
Differential pulse anodic stripping voltammetry curves recorded after 1-min electrolysis at −0.25 V in 0.1 M NaNO${}_{3}$ electrolyte solution using AAO membranes and flat aluminium oxide (AO) containing various amount of copper ions. DPASV curves are presented with the offset of 0.5 μA.

**Figure 5 ijms-23-08327-f005:**
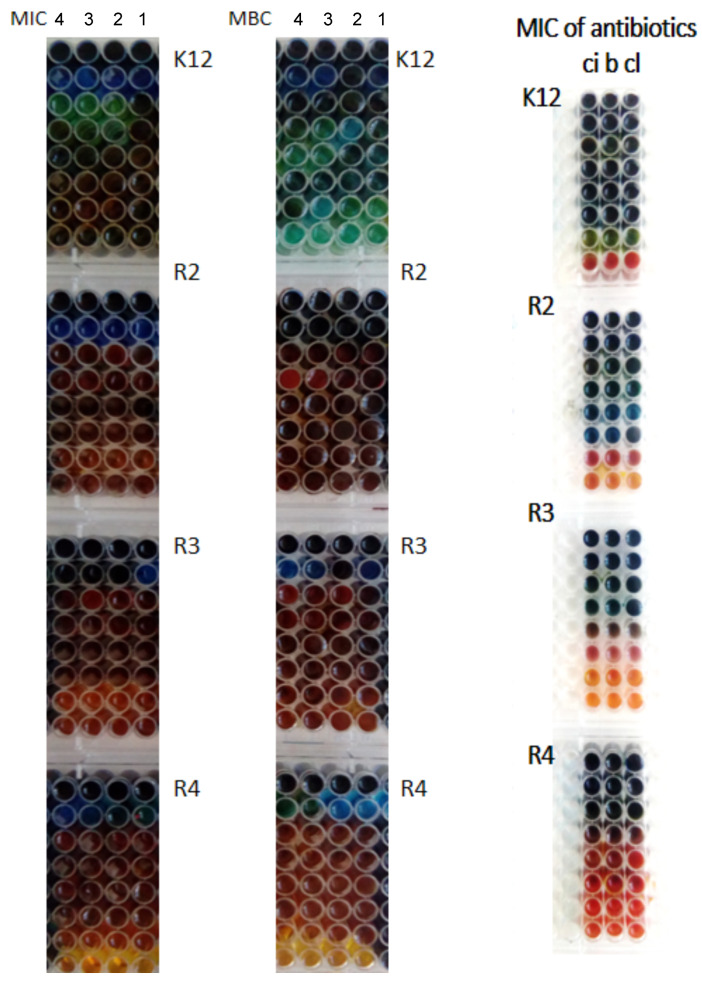
Examples of MIC and MBC on microplates with different concentrations of studied compounds (μg/cm${}^{3}$). The numbers correspond to sample names after “Cu” (“4”—AAO Cu 4). Resazurin was added as an indicator of microbial growth with K12, R2, R3, and R4 strains with the four tested compounds as described in Table 1. The third row shows examples of MIC with different strains K12, R2, R3, and R4 of the studied antibiotics: ciprofloxacin (ci), bleomycin (b) and cloxacillin (cl).

**Figure 6 ijms-23-08327-f006:**
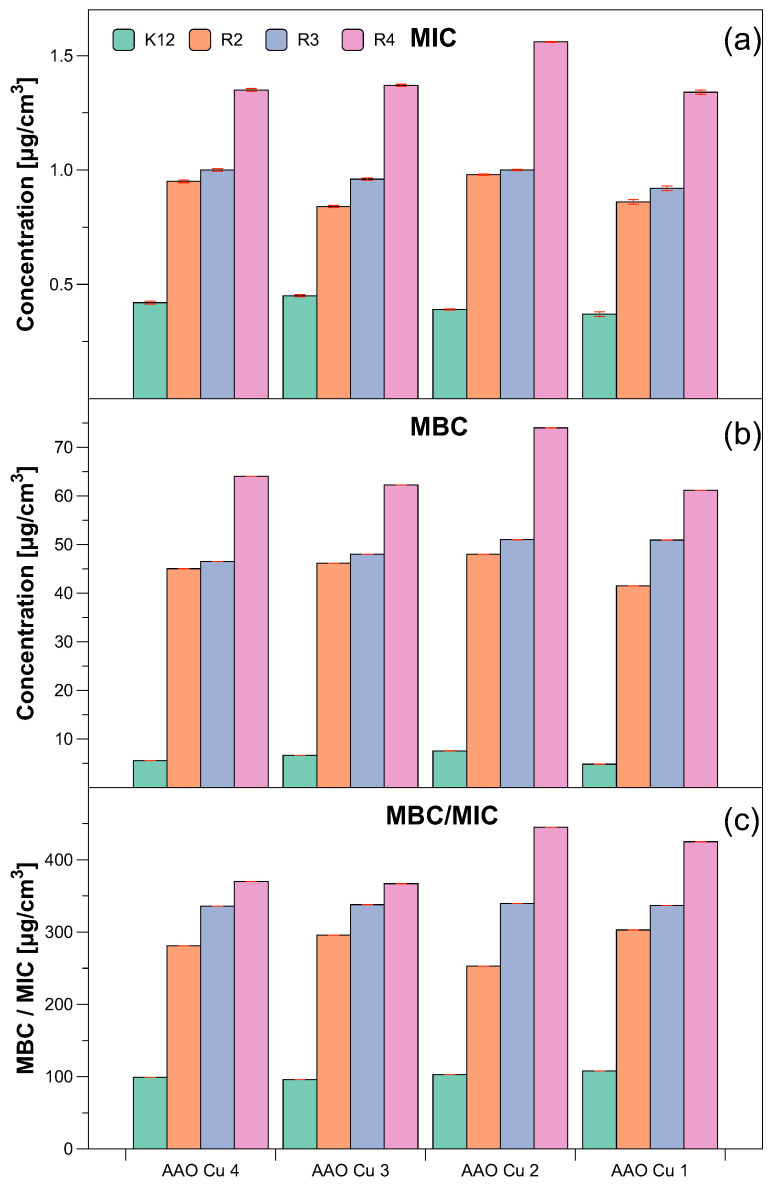
The antibacterial properties of the functionalised AAO membranes: (**a**) minimal inhibitory concentration of the AAO Cu in model bacterial strains, (**b**) minimal bactericidal concentration of the amidoximes in model bacterial strains, (**c**) MBC/MIC of the coumarin derivatives. Each experiment was performed independently in three replications (*n* = 3).

**Figure 7 ijms-23-08327-f007:**
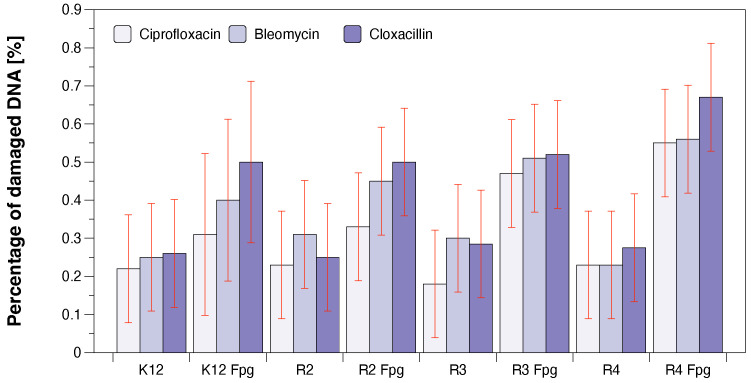
The percentage of bacterial DNA recognised by Fpg enzyme in model bacterial strains after ciprofloxacin, bleomycin, and cloxacillin treatment. The compounds were statistically significant at *p* < 0.05. Each experiment was performed independently in three replications (*n* = 3).

**Figure 8 ijms-23-08327-f008:**
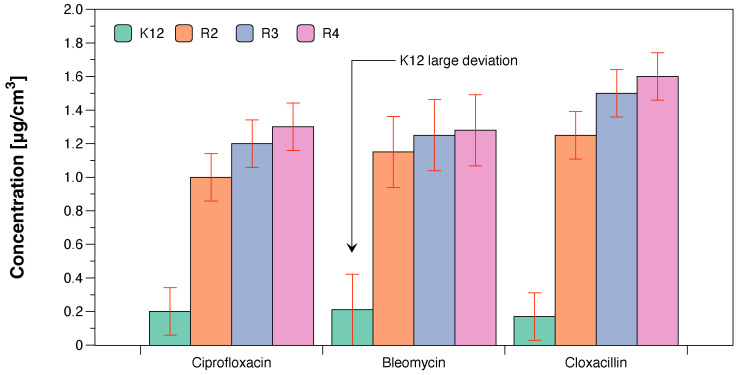
Examples of MIC with model bacterial strains K12, R2, R3, and R4 with ciprofloxacin, bleomycin, and cloxacillin. Each experiment was performed independently in three replications (n=3).

**Figure 9 ijms-23-08327-f009:**
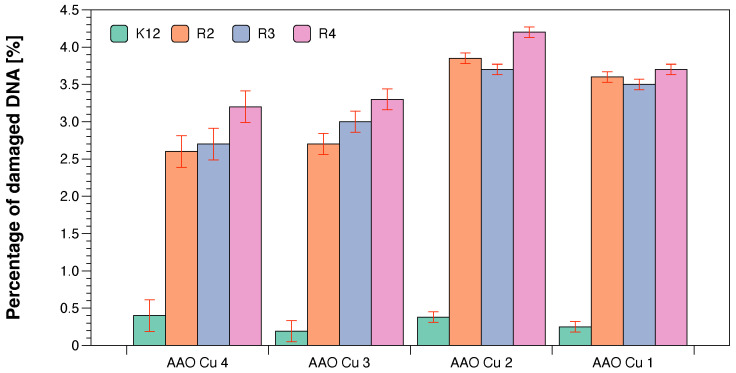
Percentage of plasmid DNA recognised by Fpg enzyme (*y*-axis) with model bacterial, K12, and R2–R4 strains (*x*-axis). Each experiment was performed independently in three replications (n=3).

**Table 1 ijms-23-08327-t001:** The statistical analysis of all measured compounds by MIC, MBC and MBC/MIC; <0.05 *, <0.01 **.

Strain	AAO Cu 4	AAO Cu 3	AAO Cu 2	AAO Cu 1	Type of Test
K12	*	*	*	**	MIC
R2	*	*	*	**	MIC
R3	*	*	*	**	MIC
R4	*	*	*	**	MIC
K12	*	*	**	*	MBC
R2	**	*	**	*	MBC
R3	**	*	**	*	MBC
R4	**	*	**	*	MBC
K12	*	*	*	*	MBC/MIC
R2	*	*	*	*	MBC/MIC
R3	*	*	*	*	MBC/MIC
R4	*	*	*	*	MBC/MIC

## Data Availability

Not applicable.

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
