# Peer review of "Functionalised Anodised Aluminium Oxide as a Biocidal Agent"

_ijms, 2022, doi:10.3390/ijms23158327_

Round 1

Reviewer 1 Report

The authors investigate anodized aluminum oxide (AAO) porous membranes with antibacterial properties. Propyl-copper-phosphonate is used to fill the membrane channels as functional units for biomedical applications. Such membranes are proposed as alternatives to polymer and silicon membranes. The results show much stronger antibacterial properties in comparison to the antibiotics ciprofloxacin, bleomycin and cloxacillin.

I have the following suggestions for improvement of the paper:

1.       The purity of the purchased chemicals should be specified. Usually this information is present on the vial label.

2.       The pore size distribution of AAO membranes could be given in case it is available from ImageJ program.

3.       The reader could have difficulty to figure out the size of the membrane plate. Is it several millimeters? The scheme on Figure 1 could be appended with the size of the matrix. Currently only the pore size of 40nm is given.

4.       The highly promising research on antibacterial and antivirus drug delivery systems based on designed peptides is not mentioned in the introduction or in the discussion. A recent paper giving an overview on designed peptides could be referred: Prediction of Amphiphilic Cell-Penetrating Peptide Building Blocks from Protein-Derived Amino Acid Sequences for Engineering of Drug Delivery Nanoassemblies, J.Phys. Chem. B 2020, 124, 20, 4069–4078

Author Response

List of revisions

and authors reply to the reviewer 1 comments

Functionalised Anodised Aluminium Oxide as a Biocidal Agent

Mateusz Schabikowski, Magdalena Laskowska, Paweł Kowalczyk, Andrii Fedorchuk, Emma Szori-Dorogházi , Zoltán Németh, Dominika Kuźma, Barbara Gawdzik, Aleksandra Wypych, Karol Kramkowski and Łukasz Laskowski

We would like to express sincere thanks to the reviewer for her/his careful reviews and fruitful comments concerning our paper. We have revised our article and corrected it according to reviewer suggestions. For the reviewer's convenience, we have marked all major changes in the article. Our replies to the reviewer's comments and the list of revisions are presented below.

  1. Reviewer wrote:

The purity of the purchased chemicals should be specified. Usually this information is present on the vial label.

Response:

Thank you for the suggestion. We have indicated the purity of the used reagents according to your suggestion.

  1. Reviewer wrote:

The pore size distribution of AAO membranes could be given in case it is available from ImageJ program.

Response:

Thank you for this remark. We have added an additional figure (Figure 3) with the said distribution of the size of pores.

  1. Reviewer wrote:

The reader could have difficulty to figure out the size of the membrane plate. Is it several millimeters? The scheme on Figure 1 could be appended with the size of the matrix. Currently only the pore size of 40nm is given.

Response:

Thank you for this remark. We added the information according to your suggestion.

  1. Reviewer wrote:

The highly promising research on antibacterial and antivirus drug delivery systems based on designed peptides is not mentioned in the introduction or in the discussion. A recent paper giving an overview on designed peptides could be referred: Prediction of Amphiphilic Cell-Penetrating Peptide Building Blocks from Protein-Derived Amino Acid Sequences for Engineering of Drug Delivery Nanoassemblies, J.Phys. Chem. B 2020, 124, 20, 4069–4078.

Response:

Thank you for pointing out this valuable publication. We studied it and added as one of the references.

Reviewer 2 Report

Please see attached a few recommendations for the consideration of the authors

Author Response

List of revisions

and authors reply to the reviewer 2 comments

Functionalised Anodised Aluminium Oxide as a Biocidal Agent

Mateusz Schabikowski, Magdalena Laskowska, Paweł Kowalczyk, Andrii Fedorchuk, Emma Szori-Dorogházi , Zoltán Németh, Dominika Kuźma, Barbara Gawdzik, Aleksandra Wypych, Karol Kramkowski and Łukasz Laskowski

We would like to express sincere thanks to the reviewer for her/his careful reviews and fruitful comments concerning our paper. We have revised our article and corrected it according to reviewer suggestions. For the reviewer's convenience, we have marked all major changes in the article. Our replies to the reviewer's comments and the list of revisions are presented below.

  1. Reviewer wrote:

I suggest you make a quick reference of any existing application of antimicrobial nanomaterials.

Response:

Thank you very much for this remark. We did it according to your suggestion.

  1. Reviewer wrote:

You could move the general info about E.coli from section 3.3. (line 208-214) in the introduction to justify why investigation focus on these bacterial species

Response:

Thank you very much for this comment. We did it according to your suggestion.

  1. Reviewer wrote:

Paragraph [2.1-preparation of AAO membranes]: Please include a literature on the described procedure (unless method developed in this laboratory/research team....if so, please note explicitly).

Response:

Thank you very much for this comment. We added the information according to your suggestion.

  1. Reviewer wrote:

At the end of this section please add a paragraph with some info on the

statistical analysis performed for the presentation of these research results (e.g. ANOVA test for mean’s comparison, software used?).

Response:

Thank you for this remark. We added the information according to your suggestion.

  1. Reviewer wrote:

Figures are clear and well-designed but the legends should also indicate how results presented (mean value +- SD)?). I would suggest to indicate statistically significant differences/ order of antibacterial activity by use of letters-for the different treatments (such as: a>b>c...?)..

Response:

In the statistical interpretation of our results, the standard deviations are given in the charts, while the statistical significance in our opinion is well presented in Table 1 where the way of their presentation seemed more clear to us.

  1. Reviewer wrote:

Line 107: “Still” rather than “Sill”?

Response:

Thank you for this remark. We have corrected this.

  1. Reviewer wrote:

References Please ensure consistency when referring to the journals (in certain cases you give abbreviations but in other not

Response:

This omission was actually a wrongly inserted reference: the reference with abbreviated “journal name” in the bibliography is a book and we accidentally posted is as a journal article. This was corrected in the current version.
